# GlyMan: Glycemic Management using Patient-Centric Counterfactuals

Asiful Arefeen[1,2], Saman Khamesian[1,2], Maria Adela Grando[1], Bithika Thompson[3] and Hassan Ghasemzadeh[1]

*Abstract*—Prolonged and frequent exposure to elevated blood glucose levels (hyperglycemia) significantly increases the likelihood of developing chronic complications, such as neuropathy, nephropathy, and cardiovascular disease, along with acute symptoms like fatigue and blurry vision. While current technologies, such as continuous subcutaneous insulin infusion (CSII) and continuous glucose monitors (CGMs), can forecast adverse events like hypoglycemia and deliver small insulin doses to counteract hyperglycemia, progress in developing tailored AI-driven interventions remains limited, which poses a barrier to optimal diabetes care. To address this gap, we propose leveraging counterfactual explanations that guide patients in making targeted adjustments to their carbohydrate intake and insulin dosing to avoid abnormal glucose levels. We introduce *GlyMan*[4], a novel method that generates counterfactual behavioral recommendations aimed at helping patients and caregivers make small, informed changes to prevent hyperglycemia, thus substantially reducing both its frequency and duration. Additionally, GlyMan incorporates user preferences into its intervention process and ensures more customized and patient-centered guidance. We rigorously evaluated GlyMan using real-world data from 21 type 1 diabetes (T1D) patients using automated insulin delivery (AID) systems. Results indicate that GlyMan surpasses existing methods, delivering 76.6% valid explanations and 86% effectiveness when assessed against historical data.

*Index Terms*—Counterfactual explanations, Diabetes, Digital health, Endocrinology, Explainable AI, Insulin pump, Wearable sensors

## I. INTRODUCTION

Postprandial hyperglycemia, defined as blood glucose levels above 180 mg/dl (10 mmol/l) two hours after a meal, affects 22% to 46% of non-critically ill hospitalized patients [1]. Chronic hyperglycemia can lead to severe complications, including retinopathy, kidney failure, neuropathy, and cardiovascular diseases [2]. Additionally, diabetes imposes a high economic burden, with an average annual treatment cost of $12,022 per patient and an estimated total cost of $412.9 billion in the U.S. in 2022 [3]. These costs can be mitigated through behavioral modifications, such as maintaining a proper diet [4] and adhering to optimal medication.

For individuals with type 1 diabetes (T1D), insulin therapy is essential due to the body's inability to produce insulin. Managing optimal insulin dosing is complex and requires constant decisions regarding food intake and insulin administration. Therefore, despite advancements in automated insulin delivery (AID) systems, few T1D patients achieve recommended glycemic targets. While AI offers potential to improve glycemic control, its use in predicting and preventing hyperglycemia and hypoglycemia in T1D patients on AID systems remains limited. AI driven interventions to target dysglycemia have potential to improve glycemic control and reduce the burden of disease in patients with T1D [5], [6].

Explainable AI (XAI) presents an opportunity to design better interventions by providing transparency in model decision-making [7]. Traditional XAI techniques, such as LIME, TIME, and SHAP, primarily focus on ranking important features. Feature relevance has proved to be important in building trust in a model that works with computer vision [8], time-series [9] or even tabular data [10] but may lack actionable insight for behavioral interventions. For instance, while understanding the relevance of features in diabetic retinopathy prediction can be useful in healthcare [9], practical interventions in digital health require more precise and actionable information. Often, these explanations are provided in view of low-level features that are hardly understandable from a human perspective, which undermines the main objective of XAI.

Counterfactual explanations (CFs) are more effective for generating actionable insights, as they suggest the smallest feature changes needed to achieve a desired outcome. For example, a CF might suggest preventing hyperglycemia by reducing carbohydrate intake or delaying a meal until blood glucose reaches a certain threshold. CFs can either be derived from training data [11] or generated using adversarial techniques [12]. Studies have demonstrated CFs' potential in diabetes management, improving fasting blood sugar, systolic blood pressure, and other health metrics [13]. However, most CF applications in diabetes research overlook patient preferences, often leading to unrealistic or infeasible recommendations.

In this regard, our solution, GlyMan, stands out by incorporating stakeholders' (e.g., patients, physicians) preferences, such as feature importance, into the CF generation process. This ensures that patients are playing a role in the interventions by specifying certain behaviors should remain unchanged. GlyMan's key contributions include:

- A novel, model-agnostic, patient-centric algorithm that generates CF-based interventions to prevent postprandial hyperglycemia through behavior changes (e.g., adjusting meal timing, carbohydrate intake, insulin dosage).
- Personalized interventions that respect individual preferences and constraints.

---

[1]College of Health Solutions, Arizona State University, Phoenix, AZ 85004, USA

[2]School of Computing and Augmented Intelligence, Arizona State University, Tempe, AZ 85281, USA

[3]Department of Endocrinology, Mayo Clinic Arizona, Scottsdale, AZ 85259, USA

email: aarefeen@asu.edu

[4]code available at: https://github.com/Arefeen06088/GlyMan

- Extensive testing using a real-world clinical dataset, with competitive performance compared to existing methods based on standard validation metrics.

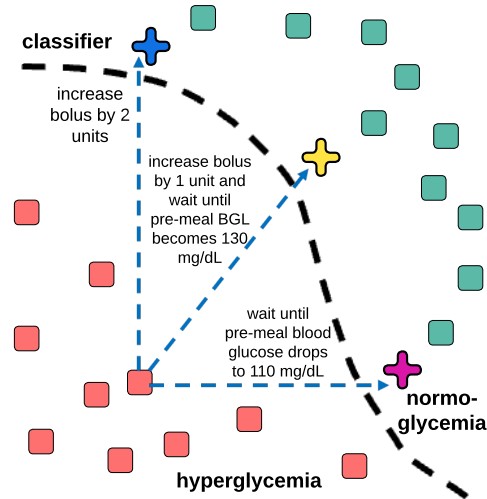

Fig. 1: Counterfactual XAI for hyperglycemia prevention.

## II. GLYMAN DESIGN

Assume that $\mathcal{D} = \{(X_1, y_1), (X_2, y_2), \dots, (X_n, y_n)\}$ be a dataset of $n$ instances that has longitudinal health observations related to eating events and the corresponding health outcome such as blood glucose level categories. Each instance $X_i = [x_i^1, x_i^2, \dots, x_i^d]$ consists of $d$ features including actionable behavioral parameters (e.g., diet, medication) and non-actionable parameters (e.g., age, gender, A1C). Considering $c$ possible classes for health outcome $Y$, where $y_i \in [1, c]$, a probabilistic AI model or classifier $f$ can be trained to map the $d$-dimensional input features to the $c$ classes and give us their corresponding prediction probabilities $f_1, f_2, \dots, f_c$:

$$f : \mathbb{R}^d \rightarrow [1, c]$$

For a test sample $X_T$ predicted to result in post-prandial hyperglycemia (i.e., $\arg\max f(X_T) = hyperglycemia$), GlyMan aims to create an effective intervention plan that guides the patient in making informed behavioral adjustments to prevent the predicted hyperglycemia, while also respecting their personal preferences.

To generate such CFs, we have to satisfy several constrains within an optimization process. For example, the CFs must belong to the desired class, must not change too much from the factuals and must reflect user preferences. We assume that the stakeholder's preferences for behavior changes are represented in vector $R(X_T) = \{r_1, \dots, r_d\}$, where each $r_i \in [0, 1]$ represents the relative importance of the $i$-th feature for modification during intervention. Specifically, a value of $r_i = 1$ indicates that the stakeholders strongly favor modifying the $i$-th feature, while $r_i = 0$ implies no preference for modification.

We formulate the CF generation process using a multi-objective optimization problem as shown in Equation (1), where the aforementioned requirements are formalized in the first to third terms, respectively.

$$\min_{X_T^*} \left[ CE\big(f_n(X_T^*), \overrightarrow{n}\big) + R \odot |X_T^* - X_T| + d\big(X_T^*, X\big) \right]$$
(1)

Here, $CE(\cdot)$ is the crossentropy loss between model's prediction on the CF and normoglycemia, $d(\cdot)$ is the distance function.

To tackle the optimization problem (1) through adversarial perturbation, we use an iterative process that modifies the features of $X_T$ incrementally by $\delta$. The adjustments are guided by saliency scores, stakeholder preferences, and the requirement to maintain realistic changes within bounds.

A primary goal of CF generation is to introduce minimal changes to the original samples. Feature saliency measures how altering a particular feature affects the model's prediction for the target class. Identifying the most salient feature during each iteration allows GlyMan to determine which feature, when adjusted, will have the greatest impact on shifting the prediction toward the desired outcome. This is done by computing the forward derivative of the model's prediction with respect to each feature.

For each modifiable feature in $x_{mod}$, the saliency score $S(x_T, y', i)$ is calculated by perturbing the feature value by a small amount and observing the change in the model's prediction probability for the target class. This change is captured through the forward derivative of the prediction with respect to the feature,

$$S(x_T, y', i) = \frac{f_n(x_T^{*i} + \delta_i) - f_n(x_T^{*i})}{\delta_i} \quad \forall x_T^{*i} \in x_{mod} \quad (2)$$

By combining feature saliency with stakeholder preference weights, a composite score is calculated to identify which feature to adjust. Specifically, the combined score $C_i$ for each feature $x_T^i$ is obtained by adding the normalized saliency score $S(x_T^*, y', i)$ (scaled to the range $[-1, 1]$) to the sum of the physician's and user's preference weights ($w_p$ and $w_u$). This score is then used to determine which feature should be modified i.e., the feature with highest composite score is selected for adjustment within the feature limits. Therefore, the index of the feature to modify is

$$i' = \arg\max_i \big[ |S(x_T^*, y', i)'| + (w_p + w_u) \big]$$

Figure 2 explains the overall framework for GlyMan.

## III. DATA

### A. Data collection

A substantial dataset was collected from 100 T1D patients treated at the endocrinology department of Mayo Clinic, Phoenix between December 2023 and April 2024 (IRB #23-003065). Each patient provided around 26 days of continuous recordings in real-world settings, including CGM data from Dexcom G6 Pro, insulin administration logs, carbohydrate intake, and device mode settings (e.g., regular, sleep, exercise)

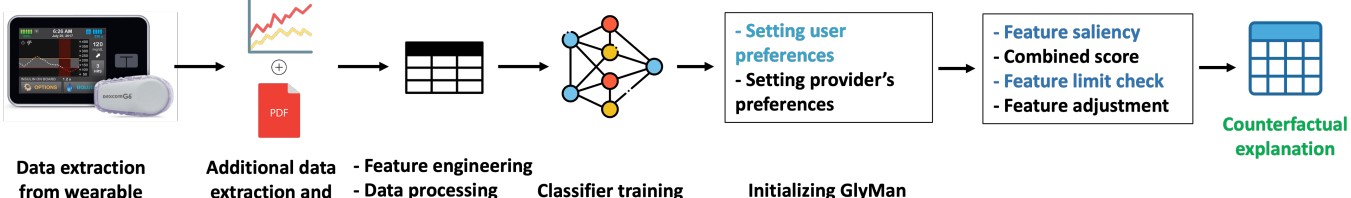

Fig. 2: Pipeline: The process begins with collecting data from T1D patients on AID technology, followed by data preprocessing and classifier training. Afterward, GlyMan is initialized with the specified preference weights and iterations are run to generate the counterfactual outcomes.

from Tandem T:SLIM X2 Pump. Of these, 21 randomly selected patients (Age: 57.4 ± 16.2 years, 11 female, A1C: 5.0-8.2%, 20 White, 1 Hispanic) contributed data that was processed further to develop and validate GlyMan.

### B. Data Processing and Feature Extraction

*1) Basal rates and device modes:* The hourly basal rates and device modes in the PDF files downloaded from Tandem are extracted by cropping the informative areas and then using an **O**ptical **C**haracter **R**ecognition (OCR) technique.

*2) Time between meal and food bolus, $\Delta t$:* Improving $\Delta t$ may play a key role in improving glycemic control. Following prior research [14], we estimated $\Delta t$ from the timeseries data

*3) Total bolus:* The total bolus is the sum of all bolus intakes taken between the time when the glucose level peaks ($t_{max}$) and the time when either the meal or the first bolus was taken, whichever occurred first, i.e., $\min(t_{meal}, t_{fb})$.

*4) Total basal:* Sum of all basal units taken between $t_{meal} - 90$ and $t_{meal}$ falls under Total basal feature.

*5) Pre-meal glucose level and slope:* The CGM reading at $t_{meal}$ is the pre-meal blood glucose level. A linear trend-line is fitted using glucose readings between $t_{meal} - 30$ and $t_{meal}$ to calculate the pre-meal slope.

*6) Filtering out carb sizes:* Patients often try to manage high blood sugar by taking extra food boluses instead of correction boluses. This can result in multiple carb intakes near the primary meal. In such cases, we consider only the largest carb size and ignore the others that occur between $\min(t_{meal}, t_{fb})$ and $t_{max}$.

Finally, we have **1361** factual samples after data curation. Two of them are depicted in Table I as examples.

## IV. EXPERIMENTAL SETUP

### A. Classifier Description

The fully-connected binary classifier for hyperglycemia classification is of 3 layers with 64, 32 and 32 neurons, respectively. Each layer has a *relu* activation and dropout rate of 0.4. The model is trained using 85% factual samples for 400 epochs with a learning rate of 0.001.

### B. Parameter Set

For preventing hyperglycemia, we set target class $y' = normoglycemia$, the corresponding prediction confidence ($\gamma$) at 0.6, maximum iterations to $N = 200$ and consider **Carb**

size, **Total bolus**, $\Delta t$, and **Pre-meal BGL** as the modifiable features. Their corresponding perturbation size, $\delta$ values are 5 grams, 0.5 unit, 5 minutes, and 10 mg/dL, respectively.

### C. Validation Metrics

We assess the CFs using some standard metrics found in the literature:

*Validity* assesses whether the produced CFs genuinely belong to the desired class. High validity indicates the technique's effectiveness in generating valid CF examples.

*Nearest Neighbor Test (NN Test)* validates the effectiveness of the CFs by comparing them against historical data to determine their likely outcomes (e.g., hyperglycemia or normoglycemia) based on past similar instances. We implement it using a k-nearest neighbor (k-NN) algorithm.

*Proximity* is the $L_2$ norm distance between $X_T$ and $X_T^*$. A low *Proximity* ensures we are making small change to the factual sample by preserving the details and not over-correcting the user.

*Sparsity* is the average number of feature changes per CF. A low sparsity ensures better user understanding of the CFs.

*Violations* quantifies how frequently non-modifiable features (e.g. age, gender, insulin etc.) are changed.

*Plausibility* estimates the fraction of explanations that fall within the feature ranges derived from the data.

### D. Simulator specifications

The simulator is an XGBoost model trained with real data to validate the CFs. With a max-depth of 13, learning rate of 0.1, 100 estimators and 85% trainig data, the XGBoost simulator achieves 80.14% accuracy.

## V. RESULTS

### A. Classifier Performance

The dense net classifier trained on the Mayo Clinic data achieves 81% prediction accuracy and 80.4% F1-score. The dataset is slightly imbalanced between the two classes.

### B. Evaluating the counterfactuals

The summary in Table II outlines the quality of CFs generated by GlyMan and NICE [11]. GlyMan achieves an average validity of 0.766, a proximity score of 0.327, 2.34 sparsity and perfect violation and plausibility scores, meaning that it does not modify features which are non-modifiable and all CFs

TABLE I: Examples of processed samples from the dataset.

| Age | Sex | Ethnicity | A1C | Carb size | Total bolus | $\Delta t$ | Mode | Total basal | Pre-meal BGL slope | Pre-meal BGL | Outcome |
|---|---|---|---|---|---|---|---|---|---|---|---|
| 61 | F | White | 6.7 | 20 | 7.57 | -5 | regular | 2.475 | 2.943 | 129 | normoglycemia |
| 32 | F | Hispanic | 5 | 35 | 5.83 | 15 | regular | 0.357 | 1.457 | 134 | hyperglycemia |

TABLE II: Evaluating the counterfactuals from GlyMan using validity, NN test, proximity, violations and plausibility.

| Method | Mayo Clinic Data | | | | | |
|---|---|---|---|---|---|---|
| | validity | NN test | proximity | sparsity | violations | plausibility |
| GlyMan | **0.766** | **0.859** | 0.327 | 2.34 | **0** | **1.0** |
| NICE [11] | 0.688 | 0.688 | **0.179** | **1.875** | 0.41 | 0.9 |

are produced within the original data manifold. However, in terms of sparsity, GlyMan underperforms by modifying more than two features per explanation on average. Additionally, the high NN test score of 0.859 suggests that subjects achieved normoglycemia in 86% of cases when encountering situations similar to the generated CFs in real life. GlyMan outperforms NICE in metrics like validity, NN test, violation counts and plausibility. NICE has a better proximity rating because it identifies CFs from the training data.

We perform several ablation studies to understand how changing certain parameters of GlyMan impacts different aspects of the produced CFs. Reducing the number of modifiable features leaves less flexibility for GlyMan to operate. In fact, it gets harder to toggle the class by modifying a fewer number of features. When GlyMan is given 4 modifiable features, it can convert all the factual samples into CFs and achieve 100% conversion rate. However, as we reduce the number of modifiable features to 1, average conversion rate drops to as low as 41%. While the conversion rate drops, GlyMan still ensures that the CFs produced are of high quality. Hence, the validity remains close to 0.75 in spite of a dip in conversion rate with just one modifiable feature. Figure 3 depicts the aforementioned analysis.

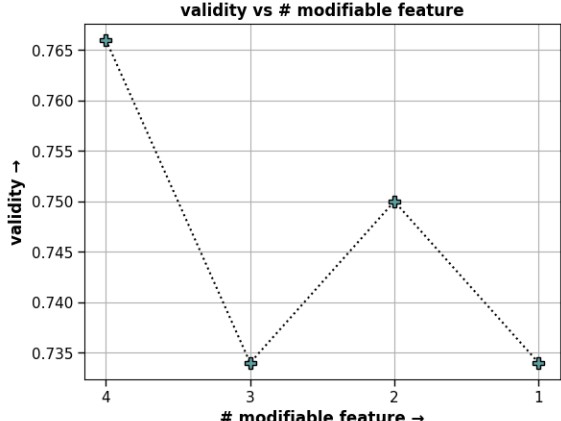

Fig. 3: Monitoring the changes in validity when the number of modifiable features is varied.

## VI. CONCLUSION

We designed GlyMan to actively involve stakeholders in the CF generation process, eliminate the need for an additional generative model, and thereby reduce training complexity. Built on real-world data from uncontrolled environments, GlyMan generates valid, fair, realistic, and minimal CFs. We evaluated and compared these CFs extensively using metrics and methods from past studies. However, GlyMan is not yet clinically validated and faces challenges such as high computational overhead due to iterative processes. Moving forward, we plan to address these limitations and trial GlyMan with real patients in a clinical setting.

## ACKNOWLEDGMENT

This work was supported by the National Institute of Diabetes and Digestive and Kidney Diseases of the National Institutes of Health under Award Number T32DK137525. Any opinions, findings, conclusions, or recommendations expressed in this material are those of the authors and do not necessarily reflect the views of the funding organization.

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
