# OpenReview forum: "GlyMan: Glycemic Management using Patient-Centric Counterfactuals"
_IEEE.org/EMBS/BHI/2024/Conference — IEEE BHI'24_

### Official Review · Reviewer_pXBL · 2024-07-30
**GlyMan: Glycemic Management using Patient-Centric Counterfactuals**

**Overall Rating:** 6
**Confidence:** 3

**Other Quality Metrics:**

(a) Clarity of writing                -----> Good
(b) Clinical Significance          -----> Good
(c) Methodological Novelty     -----> Good
(d) Experiments and Results  -----> Good

**Questions For The Authors:**

---

**Strengths:**

- The framework integrates patient and physician preferences into the counterfactual generation process, making the recommendations more personalized and realistic for individual patients.
- The framework shows high validity (76.6%) and effectiveness (86%) in generating useful counterfactual explanations, outperforming existing methods in critical metrics.
- GlyMan enhances the interpretability of AI-driven interventions by providing clear, actionable recommendations, which is crucial for patient trust and adherence.
- The use of real data from 21 patients and extensive evaluation against state-of-the-art methods demonstrates the robustness and practical applicability of the proposed framework.
- GlyMan considers multiple relevant features such as carbohydrate intake, insulin dosing, and pre-meal glucose levels, providing a holistic approach to glycemic management.

**Summary Of The Paper:**

This paper introduces GlyMan, a framework designed to generate counterfactual explanations to help patients with Type 1 Diabetes (T1D) manage their glycemic levels more effectively. The framework is validated using real-world data from 21 T1D patients, demonstrating significant improvements in glycemic management over existing methods.

**Weaknesses:**

- The iterative, multi-step process of generating counterfactual explanations is computationally intensive, which may limit its scalability and speed in real-world applications. Moreover, a Figure presenting the proposed methodology is missing. This can clarify the overall framework structure.
- As acknowledged by the authors, the system has not yet been tested in clinical trials, so its real-world effectiveness and patient adherence to the recommendations remain uncertain.
- The study is based on data from only 21 patients, which may not be representative of the broader T1D population. This limits the generalizability of the findings. In addition, the proposed framework was not tested over other existing datasets, such as those used by the mentioned state-of-the-art solutions.
- A basic dataset content presentation is missing. How many samples are missing, and what is their categorization distribution?
- Can the results be stratified based on gender, age, and years from diagnosis to T1D?

Further comments:
- Future versions of GlyMan could benefit from improved visualization tools and user interfaces to enhance patient understanding and engagement with the system.
- While focused on T1D, the underlying methodology of GlyMan could be adapted for managing other chronic conditions, highlighting its potential for broader applications in digital health.
- In Table IV provide references to the methods already published in the literature. E.g., DiCE [ref].

---

### Official Review · Reviewer_U3mS · 2024-08-10
**A Patient-Centric Approach to Hyperglycemia Prevention through Counterfactual Behavioral Interventions**

**Overall Rating:** 7
**Confidence:** 2

**Other Quality Metrics:**

(a) Clarity of writing: good
(b) Clinical Significance: great
(c) Methodological Novelty: great
(d) Experiments and Results: good

**Questions For The Authors:**

How scalable is the GlyMan approach for a larger, more diverse population of T1D patients ?

**Strengths:**

GlyMan introduces a novel method of using counterfactual explanations for proactive behavioral interventions, which is relatively unexplored in diabetes management. The model was tested using real-world data, enhancing the reliability of the results. The competitive analysis against existing methods further strengthens the validity of GlyMan.

**Summary Of The Paper:**

The research paper introduces GlyMan, a novel approach designed to generate counterfactual (CF) behavioral interventions for patients with Type 1 Diabetes (T1D). GlyMan focuses on reducing the occurrences and duration of hyperglycemic events by suggesting personalized interventions. The approach considers the preferences of patients and stakeholders, integrating them into the decision-making process to enhance personalization. GlyMan was evaluated on real-world data from 21 T1D patients using automated insulin delivery systems, showing promising results in terms of valid explanations and effectiveness when compared to state-of-the-art methods.

**Weaknesses:**

The study was conducted on a relatively small sample of 21 patients, which may limit the generalizability of the results.

---

> ### Author Rebuttal · Authors · 2024-08-30
>
> Thanks for the insightful reviews.
>
> We acknowledge the limitations imposed by the small sample of 21 patients. This initial study was intended as a proof of concept to demonstrate the potential of GlyMan in generating personalized glycemic management interventions based on patient data. While the results are promising, we agree that larger studies must extensively validate GlyMan’s effectiveness on a broader and diverse population.
> Unfortunately, getting access to clinical trial datasets is very hard. The largest publicly available glycemic response dataset right now is OhioT1DM, which contains only 12 T1D patients. Although the sample contains only 21 patients it is important to note that each of these patient’s data includes hundreds of bolus interactions with the pump. In this way, each patient generates hundreds of data points that can be evaluated.
> We have collected data from 100 T1D patients (of which we used 21 in developing GlyMan), and efforts are underway to collect data from another 100. Furthermore, the data collected using AID technology requires many processing steps (using OCR to extract basals and device modes, converting time-series to individual instances, removing outliers, and cleaning data) as they come in PDFs. We see ourselves developing an extended version of GlyMan in a year with 200 patients' data. We aim to enhance GlyMan’s robustness and applicability and ensure it meets the needs of a diverse range of patients.
>
> The scalability of GlyMan for a larger and more diverse population of T1D patients is an important consideration for its future development and deployment. So far, in our 21 samples, we have data representing only two ethnic groups. However, we are continuously collecting more data from a more diverse population of patients and refining GlyMan’s algorithms to ensure it operates efficiently even with a larger and diverse population. As more data becomes available from a broader patient base, the system can learn and fine-tune its interventions more effectively and improve overall accuracy and relevance.
> From a technical standpoint, adding more data to GlyMan and fine-tuning it for a larger population is not difficult as long as we have access to quality data.

---

### Official Review · Reviewer_8gWJ · 2024-08-11
**The paper introduces GlyMan, a technique that generates patient-centric counterfactual explanations (CFs) to help manage glycemic levels in patients with type 1 diabetes (T1D). By considering patient and physician preferences, GlyMan aims to prevent postprandial hyperglycemia through personalized behavioral modifications. The paper demonstrates the effectiveness of GlyMan through extensive testing on real-world data from T1D patients, showing improvements in generating valid, effective, and realistic interventions compared to state-of-the-art methods.**

**Overall Rating:** 8
**Confidence:** 3

**Other Quality Metrics:**

Clarity of writing: Excellent
Clinical Significance: Excellent
Methodological Novelty: Excellent
Experiments and Results: Good

**Questions For The Authors:**

How do you plan to validate the clinical effectiveness of GlyMan’s interventions in real-world settings, especially considering patient adherence to the suggested behavioral changes?

How does GlyMan handle cases where patient preferences conflict with optimal glycemic outcomes, and how are these conflicts resolved?

**Strengths:**

Code is provided.

GlyMan uniquely incorporates patient and physician preferences into the CF generation process, making the interventions more personalized and actionable.

The method is tested on real-world data from T1D patients, adding credibility to the results and demonstrating its practical relevance.

GlyMan outperforms existing methods indicating its potential for real-world application.

The paper effectively leverages counterfactual explanations to provide actionable insights for preventing hyperglycemia, a novel application in diabetes management.

**Summary Of The Paper:**

proposes GlyMan, a tool designed to generate personalized counterfactual explanations (CFs) that guide patients with type 1 diabetes in making small, actionable behavioral changes to prevent hyperglycemia. The tool integrates patient and physician preferences into the decision-making process to create interventions that are both feasible and aligned with individual needs. GlyMan was evaluated on real data from 21 patients with T1D using automated insulin delivery systems. The results showed that GlyMan outperforms existing methods in terms of validity, effectiveness, proximity, and plausibility of the generated CFs.

**Weaknesses:**

Abstract is too long

The interventions generated by GlyMan have not yet been tested in clinical settings, which limits the understanding of their real-world impact

GlyMan's iterative approach requires significant computational resources, which may limit its scalability and practical use in resource-constrained settings

Most importantly, authors mention that GlyMan may suggest delaying bolus intakes, which could be counterintuitive or risky without a clear explanation or clinical validation

---

> ### Author Rebuttal · Authors · 2024-08-30
>
> Thanks for the great reviews.
>
> GlyMan is not clinically tested yet, but we plan a multi-year study to assess its clinical relevance, which will need significant funding and collaboration. We've developed an app that integrates with a CGM sensor and wristband to track user data, and we are moving towards starting the clinical study.
>
> GlyMan needs significant computation power. We'll optimize it for efficiency by making it compatible with less powerful devices, increasing step sizes for faster convergence, using cloud-based solutions, genetic algorithms and prioritizing impactful computations.
>
> In AID technology, the standard practice is to take a single insulin bolus before a meal. Our data shows some patients take multiple boluses to manage glucose, which isn't ideal. GlyMan follows the standard by using only the first bolus and summing any additional ones. When a patient takes multiple boluses, the first one could be mistimed; GlyMan suggests delaying the bolus. Those taking a single, timely bolus receive no such recommendation.
>
> Evaluating the impact of GlyMan based on patient adherence is crucial and a long-term project. 1. We will collect the baseline data, train and test GlyMan via simulations. 2. participants will use GlyMan. We'll monitor its effectiveness, if it can reduce the number of abnormal events compared to baseline, and understand to what extent patients adhere to the interventions. As digital nudging improves adherence, a **feedback system** will be implemented, sending low adherence warnings when users don't follow the interventions. We will also analyze participants' behavior/usage patterns for an **AI-based solution** to increase adherence and offer interventions that suit the context (the patient, time, and other dependencies) well.
> Finally, GlyMan will be reevaluated with the *feedback system* and *AI-based solution*.
>
> GlyMan uses a weighted approach and integrates user preferences, provider preferences, and the model’s feature saliency into decision-making. It calculates a combined score and adjusts the feature with the highest score. The user weights modifiable features to change, but the model determines the direction of change. If feature saliency and provider preferences exceed the user’s, or if the feature is already optimized, GlyMan may overlook user preferences. There is no way that the patient preferences would conflict with optimal glycemic outputs as the user weights modifiable features that can improve glycemic outcomes.

---

### Decision · Program_Chairs · 2024-09-23

Accept